# Small-Area Geographic and Socioeconomic Inequalities in Colorectal Cancer in Cyprus

**DOI:** 10.3390/ijerph20010341

**Published:** 2022-12-26

**Authors:** Konstantinos Giannakou, Demetris Lamnisos

**Affiliations:** Department of Health Sciences, School of Sciences, European University Cyprus, Nicosia 1516, Cyprus

**Keywords:** colorectal cancer, incidence, mortality, socioeconomic inequalities, geographic disparity, regional deprivation, small-area analysis, Cyprus

## Abstract

Colorectal cancer (CRC) is one of the leading causes of death and morbidity worldwide. To date, the relationship between regional deprivation and CRC incidence or mortality has not been studied in the population of Cyprus. The objective of this study was to analyse the geographical variation of CRC incidence and mortality and its possible association with socioeconomic inequalities in Cyprus for the time period of 2000–2015. This is a small-area ecological study in Cyprus, with census tracts as units of spatial analysis. The incidence date, sex, age, postcode, primary site, death date in case of death, or last contact date of all alive CRC cases from 2000–2015 were obtained from the Cyprus Ministry of Health’s Health Monitoring Unit. Indirect standardisation was used to calculate the sex and age Standardise Incidence Ratios (SIRs) and Standardised Mortality Ratios (SMRs) of CRC while the smoothed values of SIRs, SMRs, and Mortality to Incidence ratio (M/I ratio) were estimated using the univariate Bayesian Poisson log-linear spatial model. To evaluate the association of CRC incidence and mortality rate with socioeconomic deprivation, we included the national socioeconomic deprivation index as a covariate variable entering in the model either as a continuous variable or as a categorical variable representing quartiles of areas with increasing levels of socioeconomic deprivation. The results showed that there are geographical areas having 15% higher SIR and SMR, with most of those areas located on the east coast of the island. We found higher M/I ratio values in the rural, remote, and less dense areas of the island, while lower rates were observed in the metropolitan areas. We also discovered an inverted U-shape pattern in CRC incidence and mortality with higher rates in the areas classified in the second quartile (Q2-areas) of the socioeconomic deprivation index and lower rates in rural, remote, and less dense areas (Q4-areas). These findings provide useful information at local and national levels and inform decisions about resource allocation to geographically targeted prevention and control plans to increase CRC screening and management.

## 1. Introduction

Cancer remains one of the leading causes of morbidity and mortality globally and an important obstacle to expanding life expectancy in every country [1]. Despite certain improvements in screening and therapy, colorectal cancer (CRC) remains the third most common malignancy globally and the second leading cause of cancer deaths, with a projected 1.9 million new cases and >930,000 deaths in 2020 [2]. Based on the projection of ageing, growth of the population, and socioeconomic development, CRC incidence is predicted to reach 3.2 million cases worldwide in 2040 [3]. Although the prospect for CRC therapy is usually positive, the rising incidence of early-onset CRC, which is commonly defined as CRC diagnosed in younger individuals (<50 years of age), poses a heavy financial burden and a huge public health challenge [4,5,6,7].

CRC development is complex, multi-factorial, and multi-mechanistic, with a wide range of genetic and non-genetic factors having an aetiologic role [8]. The increase in CRC incidence is mostly attributed to elevated exposure to modifiable environmental risk factors such as obesity, physical inactivity, poor diets, alcohol consumption, smoking, and individual and area-level socioeconomic status [6,8,9]. In contrast to individual-level socioeconomic status indicators, such as income, occupation, and education, area-level socioeconomic status reflects community characteristics [10,11]. Results have shown that area-level and individual-level socioeconomic status are two independent phenomena, both of interest, and that their separate effects as well as their interaction are crucial for understanding and reducing socioeconomic inequalities [11]. At the individual level, there was an association between education and CRC survival in men, with an excess hazard ratio in those without any education in comparison to those having some form of qualification, while in women, there is an association between occupation and CRC survival with manual/technical occupation having excess hazard ratio in comparison to intermediate and managerial/professional occupations [12]. At the area level, people living in more disadvantaged areas had lower five-year CRC survival than residents of less disadvantaged regions, and the observed survival disadvantage was greater for men than for women [13]. Moreover, there are several studies reporting an area-level socioeconomic gradient in CRC incidence [14] and indicating a higher concentration of CRC incidence among lower-income and less-educated neighbourhoods [15].

CRC is generally asymptomatic, and its symptoms appear at the advanced stages where cancers are aggressive, malignant, and metastatic [16]. Consequently, routine screening and early detection are crucial determinants for metastasis prevention, mortality reduction, and improvement of prognosis and quality of future life. Regional differences in socioeconomic status, health behaviours, and lifestyle-related factors could account for the geographical variance in CRC screening uptake. Individual socioeconomic status (e.g., lower education level, lower household income, and being unemployed) could affect CRC screening uptake as well as the quality of screening [17,18,19,20,21,22]. Lower socioeconomic status, as well as race/ethnicity, were associated with decreased access to age-appropriate screening and/or increased prevalence of behavioural risk factors for CRC [21] Area-level socioeconomic deprivation is also associated with the uptake of CRC screening which is socially graded between the most and the least deprived areas [23].

Previous studies have shown geographic disparities in CRC incidence, mortality, and survival. [24,25,26] CRC mortality disparities have been reported in Greece between rural and urban areas [27]. Studies indicate that rural areas or segregated communities face several challenges to health and well-being, including systematic disinvestment and hospital closures, while cancer screening rates are lower in such places [28,29,30]. Increased risk of mortality in rural populations may be associated with decreased screening uptake in these areas, whereas factors such as lower average educational attainment, lack of health insurance, and poverty may contribute to screening disparities [31].

In Cyprus, a high-income country and a member state of the European Union (EU), neoplasms accounted for 19% of all deaths [32]. In 2020, CRC was the third most frequent type of malignancy in Cyprus, after breast and prostate cancer, for both men and women, whereas CRC mortality rates were estimated to be higher in men than in women (age-standardised mortality rates per 100,000 is 40.4 for males and 22.9 for females) [33]. The prevalence of CRC is expected to grow in the next few decades because of the population ageing, with further imposed health burdens for the individual and healthcare-associated costs to society [34]. Knowing the geographical patterns of incidence and mortality allows health policies to be targeted at the areas of highest risk. However, the geographical disparity of CRC burden and its relationship with regional deprivation has not been studied in Cyprus. Thus, the objective of this study was to analyse the geographical variation of CRC incidence and mortality and to evaluate its association with regional deprivation in Cyprus for the time period of 2000–2015. The identification of geographical locations with greater CRC incidence and mortality rates, as well as higher degrees of deprivation, will tailor health promotion strategies and promote a targeted allocation of financial resources.

## 2. Materials and Methods

### 2.1. Data Sources and Data

The incidence date, sex, age, postcode, primary site, death date in case of death or last contact date in case of alive of all cases of CRC from 2000–2015 in Cyprus were obtained from the Health Monitoring Unit (National Cancer Registry) of the Cyprus Ministry of Health. This study used the codes C18-C20 (malignant neoplasms in the colon, rectosigmoid junction, and rectum) of the International Statistical Classification of Diseases and Related Health Problems (ICD-10. The number of CRC incidences and mortality per gender and age group (of five-year width) was calculated for each geographical area in Cyprus. There are Ν = 369 geographical areas in the Republic of Cyprus with a median population size of 316 people (IQR: 110.50–1123.50) (mean land area in square kilometre 15.41 km^2^). These geographical areas are the smallest geographical units for which both national cancer registry and census data are simultaneously available.

Indirect standardisation was implemented to measure the sex- and age-Standardised Incidence Ratios (SIRs), and Standardised Mortality Ratios (SMRs) of CRC. The national gender and age-specific CRC incidence and mortality rates were applied to the gender and age-specific population sizes of each area to compute the expected number of incidences and mortality for each area. The SIRs and SMRs for each geographical area were calculated as the ratio of observed to expected incidence and mortality, respectively, for each area.

The national small-area socioeconomic deprivation index was developed by Lamnisos et al. (2019) [35] and included four census-based area-level indicators (low education, single-person households, divorced or widowed, and single-parent households). They have used six aggregated area-level socioeconomic indicators from the 2011 national population census selected on three criteria: previous use in the development of socioeconomic deprivation indices, affinity with the material and/or social dimensions of deprivation, and availability at the municipality/community level. They used the Principal Component Analysis (PCA) to construct indicator weights, and PCA resulted in two principal components explaining 65.7% of the total variance. The first principal component included four indicators (low educational attainment, single-person households, divorced or widowed and single-parent households), and it was named “Rural-related socioeconomic deprivation” because it was related to features of deprivation that are by default more common in the rural setting of Cyprus. This deprivation index was found to have good construct and predictive validity, and it is the socioeconomic deprivation index applied in this study [35].

The study protocol has been approved by the Bioethics Committee of the European University Cyprus and has been exempt from further Bioethics approval since it is an ecological study of aggregated data at geographical level which are either publicly available (census data) or officially provided by the Health Monitoring Unit of the Cyprus Ministry of Health upon request (colorectal cancer data).

### 2.2. Statistical Methods

Moran’s I value for SMR was 0.56 (*p*-value < 0.001), and the value for SIR was 0.53 (*p*-value < 0.001), indicating significant spatial autocorrelation for those primary outcomes. Therefore, a univariate Poisson log-linear spatial model was used to include spatial dependence and to estimate smooth values of SIRs and SMRs. This statistical model assumes that the observed number of new colorectal diagnoses cases or deaths in any geographical area follows a Poisson distribution, and it is log-linearly related to the expected number of diagnosed cases or deaths in that geographical area and a random effect. The random effect is spatially structured to account for the local variability and to incorporate the propensity of neighbouring areas to show similar incidence and mortality rates. The spatially structured random effect followed the Conditional Autoregressive (CAR) model [36], with first-order adjacency defining the neighbouring areas (i.e., those areas sharing at least one common boundary). The smooth SIRs (or SMRs) in each geographical area were the ratio of the fitted number of colorectal diagnosed cases (or deaths) resulting from the spatial model to the expected number of diagnosed cases (or deaths). An SIR higher than one indicates that the specific geographical area had a higher number of cases than was expected. The smooth estimates of the Mortality to Incidence ratio (M/I ratio) for each geographical area were defined as the ratio of the fitted number of deaths to the fitted number of cases resulting both from the Poisson log-linear models.

The association of CRC incidence and mortality rate with the national socioeconomic deprivation index was examined with the Poisson log-linear spatial model. The dependent variable of this model was the number of CRC diagnosed cases (or deaths), the offset term was the expected number of CRC diagnosed cases (or deaths), and the random effect was following the CAR model with first-order adjacency. The independent or response variable was the national socioeconomic deprivation index which was entered in the model either as a continuous z-score variable or as an ordinal categorical variable with four categories corresponding to the quartiles of areas with increasing levels of socioeconomic deprivation.

The association of M/I ratio with the national socioeconomic index was investigated with the Gaussian linear spatial model. The dependent variable was the smooth M/I ratio, the independent variable was the socioeconomic deprivation index, and the random effect followed the CAR model with first-order adjacency.

The univariate Poisson log-linear and Gaussian linear spatial models were implemented following the Bayesian approach that specifies prior distributions for all the model parameters (in this case, the regression coefficients and the variance of random errors and random effects). All statistical analyses were performed in the statistical software R (R Core team, 2016) using the R package CARBayes [37] and the function S.CARLeroux. Credible intervals (Crl) of 95% were computed for the regression coefficients of the spatial models. 

## 3. Results

The CRC incidence and mortality rates per gender and age group for the time period of 2000 to 2015 are presented in Figure 1. CRC incidence rates increased with age, with more pronounced increments after the age of 50–55 years, and males had higher CRC incidence rates than females in almost all age groups. The highest CRC incidence rates were observed for males aged 80+ (48.8 per 100,000 persons), 75–79 (42.8 per 100,000 persons), and 70–74 (30.9 per 100,000 persons). The same picture emerged for the mortality rates since CRC mortality rates increase with age, and males had higher rates than females in all age groups. The highest CRC mortality rates were observed for males aged 80+ (57.1 per 100,000 persons), females aged 80+ (32.7 per 100,000 persons), and males of aged 75–79 (29.5 per 100,000) persons.

The choropleth maps of the CRC SIRs and smooth CRC SIRs are displayed in Figure 2. There were two-fold differences in the smooth CRC SIRs, with the lowest value equal to 0.67 and the highest value equal to 1.27. There were nine geographical areas with smooth SIR higher than 1.15, and most of those areas are located on the east coast of the island. 

There was also a distinctive pattern in the choropleth map of smooth CRC SIRs since eastern areas of the island had the higher incidence rates while rural, remote areas, and west coast areas had the lowest values. The same pattern was observed for CRC SMRs, with higher mortality rates observed in the eastern areas of the island and lower rates in the western areas of the island (Figure 3). 

There are five areas with mortality rates higher than 1.15, and the maximum mortality rate was 1.19. Four of those areas also have the highest CRC incidence rates. In contrast to CRC incidence and mortality, a different picture emerged in the choropleth map of the M/I ratio (Figure 4). The higher M/I ratio values were observed in the rural, remote, and less dense areas of the island, while the lower rates were observed in the metropolitan areas.

Table 1 shows the Relative Risks (RRs) of CRC incidence and mortality and Mortality-to-Incidence levels per one standard deviation (SD) increase across quartiles of areas with increasing levels of rural-related socioeconomic deprivation. There was an inverted U-shape pattern in the RRs of CRC incidence and mortality, with higher rates in geographical areas belonging to the second quartile (Q2) and lower rates in rural, remote, and less dense areas (Q4). CRC incidence appeared 13% higher in the second quartile (Q2) (95% CrI of RR: 1.01, 1.23) and 22% lower in the fourth quartile (Q4), which are the rural and remote areas (95% CrI of RR: 0.63, 0.97). CRC mortality had the same pattern, with 13% higher CRC mortality in the second quartile of areas (Q2) (95% CrI of RR: 1.01, 1.31) and 27% lower CRC mortality in the rural and remote areas (fourth quartile, Q4) (95% CrI of RR: 0.57, 0.93). In contrast to CRC incidence and mortality RRs, a different pattern emerged in the M/I ratio level with a stepwise increase across increasing levels of rural-related socioeconomic deprivation. By comparison to Q1-areas, M/I ratio levels were 0.04 higher in Q2-areas (95% CrI: 0.02, 0.06), 0.11 higher in Q3-areas (95% CrI: 0.08, 0.13) and 0.15 higher in Q4-areas (95% CrI: 0.12, 0.18).

## 4. Discussion

To the best of our knowledge, our study is the first to apply a community-based regional deprivation index to explain geographical disparities in CRC incidence and mortality in Cyprus. The findings of this ecological study, which used data from Cyprus’s National Cancer Registry, revealed that CRC incidence and mortality increased with age, and males have higher rates than females in all age categories. Our findings from the choropleth maps revealed specific geographical areas having 15% higher smooth SIR and SMR, with most of those areas located on the east coast of the island. In addition, we found higher M/I values in the rural, remote, and less dense areas of the island, while lower rates were observed in the metropolitan areas. We also discovered an inverted U-shape pattern in CRC incidence and mortality with higher rates in Q2-areas and lower rates in rural, remote, and less dense areas. Particularly, CRC incidence appeared 13% higher in Q2-areas and 22% lower in Q4-areas, which are rural and remote areas, whilst CRC mortality had the same pattern with 13% higher CRC mortality in Q2-areas and 27% lower CRC mortality in Q4-areas. A different pattern emerged in the M/I ratio level, indicating a stepwise increase across increasing levels of rural-related socioeconomic deprivation, with M/I levels being 0.04 higher in Q2-areas, 0.11 higher in Q3-areas, and 0.15 higher in Q4-areas in comparison to Q1-areas. 

The results of our study showed that CRC incidence and mortality rates are higher in males than females in almost all age groups. Concurring to our study, worldwide age-standardised incidence rates are approximately 50% higher in men than in women [38,39,40]. In line with our results, a previous study that examined the trends of CRC incidence using data from four population-based cancer registries in the Middle East, including Cyprus, found higher age-adjusted incidence rates among males [41]. The disparity in incidence rates between men and women may be due to a combination of variables, but mainly due to the higher vulnerability of men to environmental risk factors such as visceral fat, acholic beverages, smoking, and poor dietary patterns [42,43,44,45] rather than genetic factors [46]. Furthermore, compared to women, men do not benefit from the preventive impact of endogenous oestrogen in the same way as women do [47] and have lower CRC screening uptake [48,49]. Gender differences in mortality rates could also reflect the role of other factors other than environmental, which requires more research to provide conclusive evidence. We also found that CRC incidence and mortality increase with age, particularly after the age of 50. This is consistent with prior evidence, as age has been linked to a higher risk of CRC, with 90% of global cases and fatalities occurring beyond the age of 50 [2,50,51,52].

An inverted U-shape pattern in the RRs of CRC incidence with higher rates in geographical areas belonging to the second quartile of the socioeconomic deprivation index and lower rates in rural, remote, and less dense areas (Q4) was found. In particular, CRC incidence was 13% higher in the Q2-areas and 22% lower in the Q4-areas. The same pattern was observed in the RRs of CRC mortality with 13% higher CRC mortality in the Q2-areas and 27% lower CRC mortality in the Q4-areas. The research provides inconsistent findings about the relationship between CRC incidence and mortality and levels of deprivation. Previous research has linked CRC incidence and mortality to high degrees of deprivation in some studies but not in others for both men and women [52,53,54,55,56,57,58,59,60,61,62]. However, a recent systematic review and meta-analysis found that rural dwellers were significantly more likely to die from cancer than their urban counterparts (HR 1.05, 95% CI 1.02–1.07) [63]. The inconsistency in the findings of international studies may be attributed to various reasons, including differences in the definition of rural/urban areas, as well as variations in measurement, analysis, and outcomes. 

Although there is no clear evidence as to why these geographical variances exist, there are various methods by which the incidence of CRC may differ across geographic locations. Changes in regional CRC incidences could be explained by differences in screening behaviour and practices. For instance, lower screening rates as well as limited access to advanced screening and other medical facilities, have been reported in more deprived areas in various countries [54,55,64,65,66,67]. However, it is possible that other obstacles to screening, such as widespread beliefs and lack of knowledge among specific population groups, may contribute to screening uptake. Furthermore, according to a recent systematic review, being under 60 years old, obese, a smoker, and sedentary are risk factors for not participating in CRC screening programs and not visiting a doctor [68]. Regional differences in socioeconomic status, health behaviours, and lifestyle-related factors could also account for these geographical variances. Socioeconomic status (e.g., lower education level and low income) could affect CRC screening uptake as well as the quality of screening [17,18,19,20,21] and is strongly associated with smoking, alcohol consumption, and physical inactivity [21,69,70,71,72], which are risk factors for CRC. In fact, recent evidence revealed a shift in the Cypriot diet away from the traditional Mediterranean diet [73,74], and perhaps the increase in CRC is due to this increase in dietary risk factors. Of note, there is currently no national CRC screening program in Cyprus and no data for a nationwide number of colonoscopies conducted, which could explain some of the geographical differences in the study population. However, as in most registry-based analyses, due to the lack of individual socioeconomic data, we can only use speculative explanations, and thus, further investigation of these factors is needed. Disparities in access to medical and other healthcare services might be another potential cause of differences in regional CRC incidences among metropolitan and rural, remote, and less dense areas. 

In contrast to CRC incidence and mortality RRs, a different pattern emerged in the M/I ratio level, indicating a stepwise increase across increasing levels of rural-related socioeconomic deprivation with M/I ratio levels to be 0.04 higher in Q2-areas, 0.11 higher in Q3-areas and 0.15 higher in Q4-areas in comparison to Q1-areas. Survival from CRC depends heavily on the stage at diagnosis [75,76,77,78], which may explain the higher M/I ratio levels in rural, remote, and less dense areas compared to metropolitan areas. Thus, it is possible that less access to CRC screening and medical care in these areas due to poorer spatial access to healthcare may contribute to this incidence/mortality dynamic. This observation could imply that residents in more deprived areas, due to the limited access to healthcare services, are diagnosed at a more advanced stage of disease, which can affect prognosis and treatment options. In fact, a previous exploratory ecological study conducted in Cyprus identified geographical health inequalities, with rural areas suffering from higher rates of premature adult mortality [35,79]. In accordance with our findings, previous studies have reported that rural residents enter the healthcare system later and with later stages of disease than urban residents [80,81,82,83,84]. Previous research has also found that individuals living in rural areas have limited access to healthcare services [85,86,87,88,89], fewer physician visits per year, and they wait longer for CRC surgery than individuals living in urban areas [57,90,91,92]. Furthermore, a previous systematic review found that rural residence was associated with patient delay, while practitioners in rural areas were less likely to refer due to the distance from specialist services [93]. However, findings from other research studies are equivocal [94,95,96,97].

Our results contribute to growing international evidence supporting the preferred use of small area units in ecological studies, enabling reliable and comprehensive data about CRC to be collected in the general population. This study provides for the first time in Cyprus the results and tools that are necessary for accurate estimation of CRC burden and public health decisions offering important new knowledge relevant to measuring variations in CRC incidence and mortality using area-level data. Understanding the spatial distribution of CRC incidence and mortality at the small area level, as well as the regional socioeconomic inequalities associated with this distribution, may help to shape social and health policies aimed at reducing CRC mortality and social inequality in those places. Early detection is one of the most important factors in lowering CRC mortality. In this regard, our data suggest that the origin of the positive gradient for socioeconomic level may be found in either the rates of late diagnosis in the most deprived areas or in socioeconomic variations in access to optimal treatment. To diminish the socioeconomic disparities in CRC incidence, mortality, and Mortality to Incidence ratio reported in our study, regional health authorities must invest in public health initiatives to develop region-specific CRC screening, treatment planning, and management programs of CRC in certain communities in Cyprus. Health promotion strategies that address tobacco control, harmful use of alcohol, and promotion of healthy diet and lifestyle, target the same time multiple chronic diseases could be established as well as the development of education programs on prevention measures (e.g., targeted education programs for General Practitioners (GPs) in Cyprus to assist them with their patients in participating in CRC screening). These efforts could be more beneficial if they were concentrated in areas with higher mortality and/or deprivation. In addition, future efforts will be concentrated on exploring changes in geographical inequalities of CRC burden over time and, particularly, the effect of CRC screening and treatment. Moreover, the recently collected census data for 2021–2022 will be a rich source of socio-demographic compositions and socioeconomic characteristics of the areas, and this will enable us to investigate more extensively socioeconomic inequalities in the CRC burden. Finally, the development of a national Cancer Atlas will enable the investigation of interrelations between several health outcomes. Future research should also examine the rural–urban differences in cancer stage at diagnosis and treatment to elucidate the contributing factors to higher mortality in rural areas despite lower incidence rates.

Some limitations of our study must be considered. The CRC incidence and mortality rates were estimated for administrative units as defined by the Health Monitoring Unit (National Cancer Registry) of the Cyprus Ministry of Health. While the population size of these places is generally smaller than that of previous studies in the literature, they also comprise a number of big areas (municipalities) that may be heterogeneous in terms of the CRC burden of the many smaller areas that compose them [98]. This is the known modifiable area unit problem where the size and configuration of the geographical areas have a potential effect on the results of any spatial analysis [99]. In this study, we have used the smaller geographical level for which both national cancer registry and census data are simultaneously available. The smallest geographical level in Cyprus is the post-code (*n* = 793 geographical units); however, cancer registry data are not available at this geographical level. Another limitation, which is common for all deprivation indices, is that the national deprivation index was developed using decennial census data, which may not capture variations in the socioeconomic environment of several geographical areas in the period between the national census. In our case, we have used the latest available census data for the year 2011. One more limitation of the regional deprivation index is the dependency of some aggregated indicators on the sex and age distribution of the population living in the different geographical areas, such as low educational attainment, unemployment, and not owner-occupied households.

Furthermore, because this study used an ecological design, it exhibits the limitations common to all such studies, known as the ecological fallacy. Because the CRC burden and socioeconomic deprivation indicators are aggregated, the associations observed between the variables at the geographical level cannot be directly translated to the individual level [100]. However, several results have shown that area-level socioeconomic deprivation is an independent phenomenon from individual-level socioeconomic deprivation, and its effect on health, as well as its interaction with individual-level socioeconomic deprivation, are crucial for understanding and reducing socioeconomic inequalities [11]. Indeed, in the case of CRC survival, there is evidence suggesting that area-level deprivation has an independent effect on CRC survival over and above the effect of individual characteristics, but no evidence of contextual effects, that is, the associations between individual-level socioeconomic status and colorectal cancer survival was not depending on area-based deprivation [12].

Moreover, as part of its spatial analysis, this study used univariate spatial models to account for the data’s underlying spatial structure (spatial autocorrelation) and to improve precision in CRC burden estimates. A potential disadvantage of spatial smoothing is the establishment of geographic homogeneity between nearby places, which may make distinguishing different communities difficult. However, regional smoothing of CRC burden was deemed important because numerous locations have small populations, resulting in low precision in CRC burden estimates. Finally, the CRC M/I ratio does not take values on the whole range of real numbers; however, the Gaussian linear spatial model provided a reasonable fit to the data as the model residuals followed quite a reasonable normal distribution.

## 5. Conclusions

This study is the first to examine the geographical variation in CRC incidence and mortality throughout Cyprus. It shows that CRC incidence and mortality rates increase with age rapidly after the age of 50 and are higher in males than females in almost all age groups. The M/I ratio of CRC was positively associated with regional deprivation since a stepwise increase was found across increasing levels of rural-related socioeconomic deprivation. Whether these disparities can be explained by differences in cancer care could not be finally evaluated. Our findings may aid in the discovery of previously unknown high-risk zones, allowing Cyprus’s health authorities to create and implement CRC prevention management programs suited to the features of each region. The findings suggest that programs aiming at lowering the risk of CRC should prioritise socially disadvantaged communities. More research is needed to investigate the underlying causes of socioeconomic and geographical disparities in CRC detection, as well as to undertake in-depth reflection on national healthcare organisations to improve CRC detection.

## Figures and Tables

**Figure 1 ijerph-20-00341-f001:**
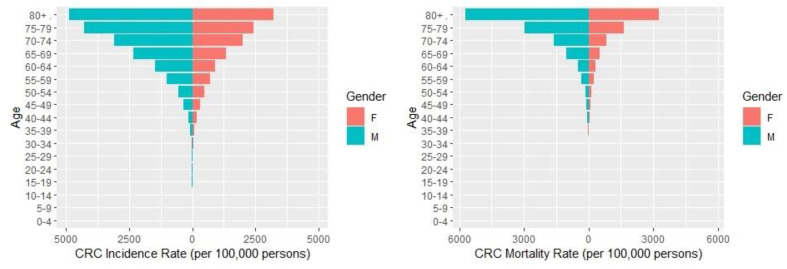
CRC incidence and mortality rates (per 100,000 persons) per gender and age group for the time period of 2000 to 2015.

**Figure 2 ijerph-20-00341-f002:**
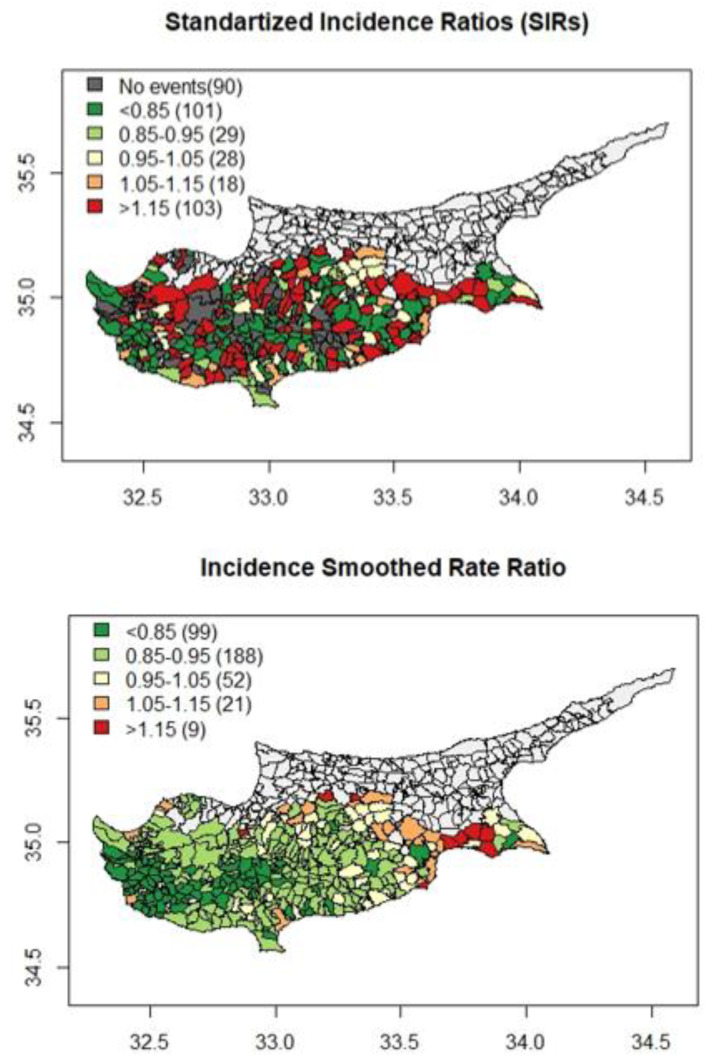
Choropleth maps of the CRC gender and age Standardised Incidence Ratios (SIRs) and smooth CRC gender and age Standardised Incidence Ratios (smooth SIRs).

**Figure 3 ijerph-20-00341-f003:**
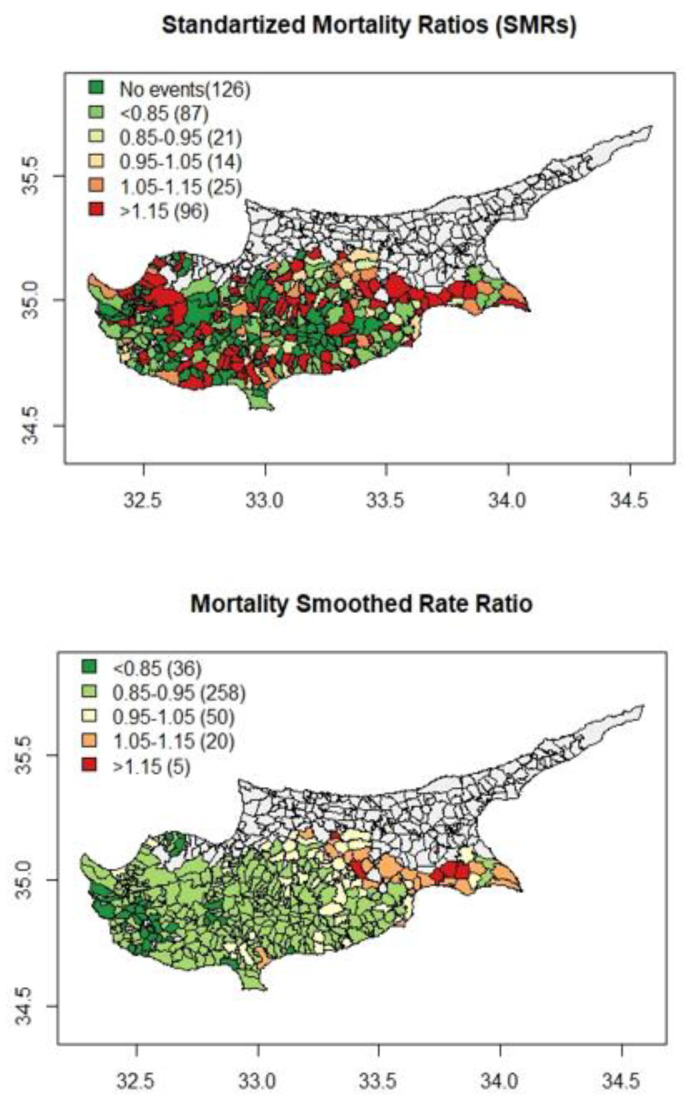
Choropleth maps of the CRC gender and age Standardised Mortality Ratios (SMRs) and smooth CRC gender and age Standardised Mortality Ratios (smooth SMRs).

**Figure 4 ijerph-20-00341-f004:**
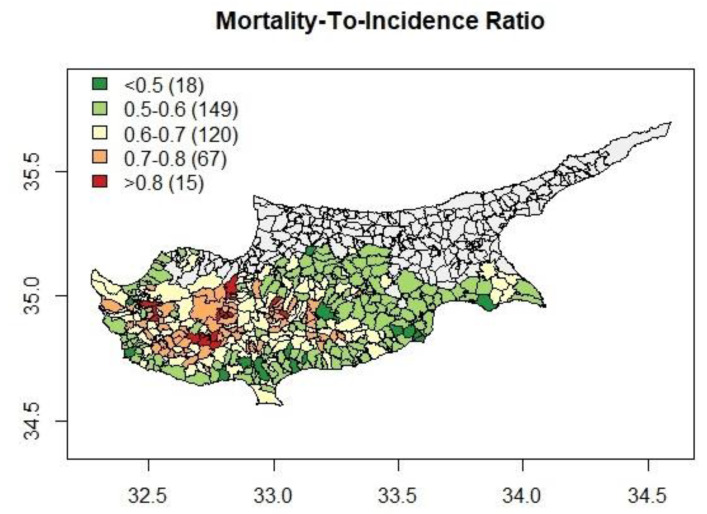
Choropleth map of the CRC Mortality-To-Incidence ratio (M/I ratio).

**Table 1 ijerph-20-00341-t001:** Relative risk (and 95% Credible Intervals) of CRC incidence and mortality and levels of CRC Mortality-To-Incidence ratio (M/I ratio) per one standard deviation (SD) increase across quartiles of socioeconomic deprivation index (*n* = 369 geographical areas).

Outcome	Per 1 SD Increase	Q1-Least Deprived	Q2	Q3	Q4-Most Deprived
CRC incidence	0.94 (0.88, 1.01)	*Ref*	**1.13 (1.01, 1.28)**	0.84 (0.70, 1.01)	**0.78 (0.63, 0.97)**
CRC mortality	0.93 (0.86, 0.99)	*Ref*	**1.13 (1.01, 1.31)**	**0.81 (0.66, 0.99)**	**0.73 (0.57, 0.93)**
CRC M/I ratio	0.06 (0.05, 0.07)	*Ref*	**0.04 (0.02, 0.06)**	**0.11 (0.08, 0.13)**	**0.15 (0.12, 0.18)**

Abbreviations: M/I ratio, Mortality-To-Incidence ratio; SD, standard deviation; CRC, colorectal cancer. Bold values indicate statistically significant associations (*p* < 0.05).

## Data Availability

The datasets used and/or analyzed during the current study are available from the corresponding author on reasonable request.

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
