# Peer review of "Small-Area Geographic and Socioeconomic Inequalities in Colorectal Cancer in Cyprus"

_ijerph, 2022, doi:10.3390/ijerph20010341_

Round 1

Reviewer 1 Report (Previous Reviewer 2)

None.

This manuscript is a resubmission of an earlier submission. The following is a list of the peer review reports and author responses from that submission.

Round 1

Reviewer 1 Report

Review of the paper “Small-area Geographic and Socioeconomic Inequalities in Colorectal Cancer in Cyprus

The paper presents results of analysis of geographical variation of colorectal cancer (CRC) incidence and mortality and explained by socio-economic indicators in Cyprus.  Overall, the paper is informative.  There are, however, some major issues which require a closer attention.   

1.      In the abstract, the authors stated “Colorectal cancer (CRC) is a leading cause of death and morbidity worldwide.”  Is this correct?

2.      In the Introduction, paragraphs 1-3 can be reduced to a single paragraph highlighting only relevant background information about CRC, and then make a smooth connection with the last paragraph of the Introduction section.

3.      The M/I ratio was treated as a normal variate. But, the support range of it is not the whole real number, which normality actually requires. Any explanation?  

A Poisson log-linear spatial model was used in the paper to account for spatial correlations. But, the authors did not provide any information how they checked the existence of a spatial correlation in order to warrant the use of such a mentioned model.

Reviewer 2 Report

1). What is the motivation behind the use of small area? What are the likely results when the geographical area is increased?

2). The use of small area is likely to lead to confounding effects. Explanation is needed.

3). The relationship between regional deprivation and cancer incidence and mortality is also widely recognized in the literature [27]. However, the geographical disparity of CRC burden and its relationship with regional deprivation has not been studied in Cyprus. Kindly survey some works related to countries having similar geography or climatic conditions with Cyprus. Countries like Turkey, Greece, Macedonia and others.

4). Line 80. Remove the full stop between literature and [27].

5). Ethical approval and statement are highly needed.

6). Map the results with works to countries having similar geography or climatic conditions with Cyprus. Countries like Turkey, Greece, and Macedonia and others. This will help in systematic review in the future.
